# Using 18F-FDG-PET/CT Metrics to Predict Survival in Ra-Dio-Iodine Refractory Thyroid Cancers

**DOI:** 10.3390/diagnostics12102381

**Published:** 2022-09-30

**Authors:** Malanie Roy, Agathe Edet-Sanson, Hervé Lefebvre, Pierre Vera, Pierre Decazes

**Affiliations:** 1Department of Nuclear Medicine, Henri Becquerel Cancer Center, 76038 Rouen, France; 2Department of Endocrinology, Diabetes and Metabolic Diseases, University Hospital of Rouen, 76000 Rouen, France; 3QuantIF-LITIS (EA[Equipe d’Accueil] 4108), Faculty of Medicine, University of Rouen, 76183 Rouen, France; 4U1239, Laboratory of Neuronal and Neuroendocrine Differentiation and Communication, INSERM, Institute for Research and Innovation in Biomedicine, 76821 Mont-Saint-Aignan, France

**Keywords:** thyroid cancer, RAI-refractory, 18F-FDG-PET/CT, prognostic factors

## Abstract

Radio-iodine refractory (RAI-R) differentiated thyroid cancer (DTC) is a rare disease with a poor prognosis and limited therapeutic resources. Therefore, identifying prognostic factors is essential in order to select patients who could benefit from an early start of treatment. The aim of this study is to identify positron emission tomography with 18F-fluorodeoxyglucose with integrated computed tomography (18F-FDG-PET/CT) parameters to predict overall survival (OS) in patients with RAI-R DTC. In this single-center retrospective study, we analyze the 18F-FDG-PET/CT parameters of 34 patients with RAI-R DTC between April 2007 and December 2019. The parameters collected are MTV, SUVmax and progression for each site of metastasis (neck, mediastinum, lungs, liver, bone) and total sites. ROC curves, Kaplan–Meier survival analysis curves, univariate and multivariate Cox analyses determine prognostic factors for 1-year and 5-year OS. The parameters for mediastinum, liver and total sites are significantly associated with worse 1-year and 5-year OS by both ROC curve analysis and Kaplan–Meier survival analysis. Univariate Cox analysis confirms significance of mediastinum SUVmax (HR 1.08; 95% CI [1.02–1.15]; *p* = 0.014) and total SUVmax (HR 1.06; 95% CI [1–1.12]; *p* = 0.042) for worse 1-year OS; of mediastinum SUVmax (HR 1.06; 95% CI [1.02–1.10]; *p* = 0.003), liver SUVmax (HR 1.04; 95% CI [1.01–1.08]; *p* = 0.02), liver MTV (HR 2.56; 95% CI [1.13–5.82]; *p* = 0.025), overall SUVmax (HR 1.05; 95% CI [1.02–1.08]; *p* = 0.001) and total MTV (HR 1.41; 95% CI [1.07–1.86]; *p* = 0.016) for worse 5-year OS. Multivariate Cox analysis confirms a significant association between liver MTV (HR 1.02; 95% CI [1–1.04]; *p* = 0.042) and decrease 1-year OS. In this study, we demonstrate that in RAI-R DTC, 18F-FDG-PET/CT parameters of the mediastinum, liver and overall tumor burden were prognostic factors of poor 1-year and 5-year OS. Identifying these criteria could allow early therapeutic intervention in order to improve patients’ survival.

## 1. Introduction

Thyroid neoplasm is a rare disease with an incidence of 586,000 worldwide and 87,000 in Europe [1]. Differentiated thyroid carcinoma (DTC) represents over 90% of all thyroid neoplasms [2], of which 10–30% are metastatic [3,4]. 

Treatment of metastatic DTC includes levothyroxine at suppressive dosage, local treatments (such as surgery, external beam therapy, thermoablation, etc.) and radio-iodine (RAI) therapy [5,6]. Unfortunately, a range of 60–70% of metastatic DTCs will become RAI refractory (RAI-R) [7]. The definition of RAI-R tumors is controversial, but most recommendations and publications retain the following criteria [8]: absence of initial RAI uptake, absence of RAI uptake after treatment with RAI in at least one metastasis, progression of at least one metastasis despite RAI uptake in all metastases and extensive RAI exposure, which includes patients who have received cumulatively 600 mCi (22.2 GBq) or more of RAI without signs of remission.

Mortality rates of patients with DTC from all stages are low (93% overall survival at 5 years) [9] but rise considerably in the case of metastasis (from 25 to 40% overall survival at 10 years) [7,10] and are even higher in the case of RAI-R metastatic DTC (10% overall survival at 10 years) [7]. Unlike most DTCs, RAI-R metastatic DTCs median survival is low, estimated at 3–5 years [11]. 

Therapeutic resources for RAI-R metastatic DTCs are limited and can significantly impair patients’ quality of life. The two multikinase inhibitors available for the treatment of RAI-R DTC in Europe, lenvatinib [12] and sorafenib [13] led to 64–85% of grade 3 or more adverse events [14]. For these reasons, for asymptomatic patients with stable or slowly progressive RAI-R metastatic DTC, a monitoring attitude is recommended, with the aim of therapeutic sparing [8]. A treatment by multikinase inhibitors should be considered in case of tumor progression and/or threatening metastasis when a focal treatment is not possible [6].

Therefore, the identification of factors of poor prognosis is essential in order to start early targeted treatment for patients who could benefit from it. Several staging systems to select patients at high risk of cancer death for more aggressive treatment have been developed, such as the EORTC system [15], MACIS system [16] or AMES score [17]. Unfortunately, none of these systems were able to accurately predict cancer-related death risk at an individual level [18]. The factors of poor prognosis identified to date are male gender [9], age≥ 55 years old [19], doubling time of thyroglobulin <1 year [20], histologic type [21], the presence of BRAF^V600E^ mutation [22], the absence of iodine uptake [7,10,23] and a high tumor burden [7,24,25].

An ^18^F-FDG PET/CT is recommended for patients with metastatic DTC in order to identify the sites of metastasis and FDG status [6]. PET-positive lesions are generally of less differentiated histologic type compared with primary thyroid tumors [26]. Several studies have shown that the intensity of FDG avidity [23], the number of FDG avid lesions [24,27] and the volume of PET-positive lesions [28] are prognostic factors of poorer survival, while the intensity of ^18^F-FDG uptake is not predictive of tumor growth in patients with metastatic DTC [29].

The aim of this study is to determine ^18^F-FDG PET/CT parameters predictive of survival in patients with RAI-R metastatic DTC.

## 2. Methods

### 2.1. Study Population

This retrospective observational study included patients treated at the Henri Becquerel Cancer Centre, Rouen, France, between April 2007 and December 2019. The study has been approved by the institutional review board. Patients were informed about the use of anonymized data for research and their right to oppose this use. The inclusion criteria were patients with (i) DTC confirmed by our pathologist; (ii) presence of neck recurrence or distant metastasis; (iii) presence of at least one radio-iodine refractory site; (iv) availability of an ^18^FDG-PET/CT performed in the year of diagnosis of RAI-R DTC; and (v) absence of systemic treatment, but radioactive iodine, or local treatment before the reference ^18^FDG-PET/CT. Data obtained from all patients included the following information: sex, age at disease onset, pathology, margin resection status, primary disease stage (8th edition of the AJCC/TNM staging system), cumulative RAI dose, site of metastasis, history of previous treatments, Tyrosine Kinase Inhibitor (TKI) treatment initiated during the 12 months after diagnosis of RAI-R, delay to death and cause of death. 

### 2.2. Radioactive Iodine-Refractory Diagnosis

The criteria we used for defining RAI-R tumors followed recent guidelines and studies [8,30,31,32]. We classified patients into the following three categories: “no RAI uptake”, which included patients with absence of initial RAI uptake or absence of RAI uptake after treatment with RAI in at least one metastasis; “disease progression despite RAI avidity” defined by the progression of at least one metastasis despite RAI uptake in all metastases; “extensive RAI exposure”, which included patients who received 600 mCi (22.2 GBq) or more of RAI cumulatively without signs of remission.

### 2.3. PET/CT Acquisition and Reconstruction

All patients underwent ^18^FDG PET/CT scans, which were performed after a 6-h fasting period and when blood glucose level was less than 1.7 g/L before the injection of the radiotracer. FDG PET/CT scans were performed under thyroxine treatment in all cases. Data from PET were acquired from the mid-thigh toward the base of the skull approximately 60 min after injection of a weight-adjusted dose of 3.0–4.5 MBq/kg. The PET system was normalized daily, and the calibration coefficient validated if the day-to-day variation remained below 0.3%. The global quantification, from the dose calibrator to the imaging system, was measured internally on a quarterly basis and double-checked by the EARLs quality assurance program. 

### 2.4. PET/CT Metrics

The manual segmentation of lesions was performed using semi-automatic software (PET VCAR, General Electrics^®^). A volume of interest was set around each lesion on the PET images. Each FDG-avid lesion was subjected to segmentation, which allowed calculation of the SUVmax (maximum standardized uptake value) and the MTV (metabolic tumor volume). The SUVmax was measured by using a volume of interest with Standardized Uptake Value (SUV) being expressed using the following definition of SUV (g/mL) = (Tissue activity (Bq/mL)/[(injected activity (Bq)/body weight (g)]). The SUVmax represents the voxel with the highest intensity of uptake in each lesion and reflects the glucose avidity of the lesion. The MTV of each lesion, representing the volume measured in the volume of interest, was determined using margin thresholds set at 42% of the SUVmax, as recommended by the European Association of Nuclear Medicine [33]. If necessary, a manual adjustment of the segmentations was performed by adapting the intensity threshold. MTV is complementary to SUVmax as it integrates information from voxels of the whole lesion. Progression of lesions was defined as the appearance of a new lesion or an increase of at least 20% of the sum of diameters of a pre-existing lesion on the reference PET/CT, in comparison with an ^18^FDG PET/CT or CT scan undertaken during the 12 previous months.

### 2.5. Statistical Analysis

Descriptive statistics of the population and results were performed with continuous variables reported as mean ± standard deviation (SD) and categorical variables as frequencies (percentage). The statistical analysis was conducted using the R software (version 3.4.2) [34]. To characterize the relationship between MTV and SUVmax, we computed the Spearman’s rank correlation coefficient between each pair of PET/CT metrics. Receiver operator characteristics (ROC) analysis was performed on continuous variables to determine prognostic value. The effect of individual parameters on OS was studied with a Kaplan–Meier (KM) survival analysis using dichotomous grouping based on the threshold determined by ROC analysis. A log-rank test was used for the comparison between survival curves. Statistical significance was considered at *p* < 0.05. Univariate, multivariate and multivariate stepwise analyses were performed using a Cox proportional-hazards model to test the relationship between the study variables and survival rates. 

## 3. Results

### 3.1. Patients’ Characteristics

Forty-four patients were included in this study, but ten were subsequently excluded as PET images were missing for analysis. Subsequently, thirty-four patients (19 women and 15 men; mean age at diagnosis of RAI-R DTC as follows: 65 years; range from 34 to 82 years) were analyzed. 

Patients were followed for a mean of 2 years and 9 months (range 2 months to 12 years). Patients’ characteristics for the whole population are described in Table 1.

Pathology was papillary in 14 cases (42%), follicular in 9 cases (26%) and poorly differentiated thyroid carcinoma in 11 cases (32%). All patients underwent total thyroidectomy with or without neck dissection as primary treatment and resection margins were complete in 23 cases (68%) and incomplete (micro or macroscopically) in 11 cases (32%). Pathology analysis revealed a vascular invasion in 20 cases (59%). At the time of initial diagnosis, DTC was classified according to the AJCC/TNM staging system as stage I for 8 patients (24%), stage II for 17 patients (50%), stage III for 2 patients (5.5%), stage IVa for 2 patients (5.5%) and stage IVb for 5 patients (15%). 

All patients received RAI administration after either thyroid hormone withdrawal or injection of recombinant thyroid stimulating hormone, with a median number of RAI treatments of 2 (range from 1 to 6). Patients were classified as RAI-R because of metastasis with no RAI uptake on a post-therapeutic scan in 27 cases (79%), because of disease progression despite RAI avidity in 5 cases (15%) and because of extensive RAI exposure in 2 cases (6%).

TKI treatment was initiated during the 12 months after the diagnosis of RAI-R DTC for 4 of the patients in our study (12%). Amongst these patients, 1 was treated with sorafenib (3%), 1 with lenvatinib (3%) and 2 with pazopanib (6%).

Metastases were located in the neck for 24 patients (71%), mediastinum for 13 patients (38%), lungs for 24 patients (71%), liver for 8 patients (24%), bone for 13 patients (38%) and in other sites for 5 patients (15%), including the brain (*n* = 2), pancreas (*n* = 2) and adrenal glands (*n* = 1). Progression during the past year was observed in 13 of the 24 neck lesions (54%), 6 of the 13 mediastinum lesions (46%), 9 of the 24 lung lesions (38%), 7 of the bone lesions (54%), 6 of the 8 liver lesions (75%) and 3 of the 5 lesions in other sites (60%).

The death occurred in 20 of the 34 patients (59%) with a median delay of 22 months (range 1 month to 8 years and 3 months). The causes of death were a respiratory deficiency in 8 cases (40%), heart failure in 2 cases (10%), kidney failure in one case (5%) and unknown in 9 cases (45%). Amongst known factors of poor prognosis, none were statistically significant in our study. The death rate was 66.7% for men vs 52.6% for women. In univariate analysis, age (HR 0.98; 95% CI [0.9–1.05]; *p* = 0.53) and pathology type (HR 1.17; 95% CI [0.2–7.03]; *p* = 0.86) were not predictive of overall survival at 1 year.

### 3.2. PET/CT Metrics and Correlation

The analyzed parameters were the metabolic tumor volume (MTV) of metastasis in each site (neck, mediastinum, lungs, liver, bone and other) and the SUVmax (maximum standardized uptake value) for each site (Figure 1). These parameters are described in Table 2. These parameters were all highly correlated by Spearman’s correlation, with a correlation coefficient close to 1 (minimal ρ = 0.74) regardless of the site of metastasis. 

### 3.3. ROC Curve Analysis

The numeric results of the ROC curve analysis for 1-year OS are shown in Table 3. These data reveal a significant decrease in survival for metastasis located in the mediastinum, liver and overall tumor burden. The MTV cut-off points were respectively 3.21 cm^3^ (AUC 0.869; *p* = 0.002) for the mediastinum, 1.16 cm^3^ (AUC 0.824; *p* = 0.001) for the liver and 79.5 cm^3^ (AUC 0.876; *p* = 0.004) for overall tumor burden. The SUVmax cut-off points were 3.9 (AUC 0.862; *p* = 0.002) for the mediastinum, 5.3 (AUC 0.824; *p* = 0.001) for the liver, and 18.61 (AUC 0.779; *p* = 0.026) for the maximum value of SUVmax. Progression in the mediastinum (AUC 0.748; *p* = 0.004) and in the liver (AUC 0.748; *p* = 0.004) was associated with a significant decrease in 1-year OS. Mediastinum SUVmax and total MTV had the best sensitivity (1.0) and liver SUVmax had the best specificity (0.897).

The numeric results of the ROC curve analysis for 5-year OS are shown in Table 4. All parameters were associated with a significant decrease in survival, except for lung MTV (*p* = 0.099) and progression (*p* = 0.437), neck progression (*p* = 0.737) and overall progression (*p* = 0.396).

### 3.4. Kaplan–Meier Survival Analysis

A Kaplan–Meier survival analysis was performed according to the cut-off value of the ROC curves. Mediastinum MTV (*p* = 0.0015), SUVmax (*p* = 0.00086) and progression (*p* = 0.006), liver MTV (*p* = 0.0015), SUVmax (*p* < 0.0001) and progression (*p* = 0.006), and total MTV (*p* = 0.00032) were found as risk factors of 1-year OS. This confirmed that these variables were robust prognostic factors at a univariate level with statistical significance. The corresponding curves are given in Figure 1.

The Kaplan–Meier analysis was also performed on values for 5-year OS, showing statistical significance for SUVmax in all metastatic sites as well as for MTV except in the lungs (*p* = 0.18).

### 3.5. Cox Univariate and Multivariate Analysis

Univariate Cox analysis showed a significant decrease in 1-year OS for neck SUVmax (HR 1.06; 95% CI [1.01–1.12]; *p* = 0.019), mediastinum SUVmax (HR 1.08; 95% CI [1.02–1.15]; *p* = 0.014), lung SUVmax (HR 1.08; 95% CI [1–1.16]; p = 0.049), total SUVmax (HR 1.06; 95% CI [1–1.12]; *p* = 0.042), mediastinum progression (HR 8.25; 95% CI [1.37–49.7]; *p* = 0.02) and liver progression (HR 8.25; 95% CI [1.37–49.7]; *p* = 0.02). These data are presented in Table 5. In binary analysis, only liver MTV (HR 15.2; *p* = 0.015) was correlated with a significant 1-year OS decrease.

Univariate Cox analysis showed a significant decrease in 5-year OS for neck MTV (HR 2.77; 95% CI [1.16–6.62]; *p* = 0.02) and mediastinum progression (HR 6.87; 95% CI [2.35–20.06]; *p* < 0.001), liver MTV (HR 2.56; 95% CI [1.13–5.82]; *p* = 0.025) and progression (HR 6.87; 95% CI [2.26–20.82]; *p* < 0.001), total MTV (HR 1.41; 95% CI [1.07–1.86]; *p* = 0.016) and SUVmax of all sites except bone, (HR 1.04; 95% CI [1–1.08]; *p* = 0.052). These data are presented in Table 6.

Univariate Cox analysis showed no correlation between TKI treatment and 1-year OS (HR 1.03; 95% CI [1.01–1.12]; *p* = 0.98) and 5-year OS (HR 1.9; 95% CI [0.76–4.87]; *p* = 0.17).

Multivariate Cox analysis showed a significant decrease in 1-year OS only for neck MTV (HR 1.04; 95% CI [1.01–1.07]; *p* = 0.015) and liver MTV (HR 1.02; 95% CI [1–1.04]; *p* = 0.0418). These data are presented in Table 7. In a multivariate stepwise Cox analysis of 1-year OS including all parameters (MTV, SUVmax and progression), only mediastinum SUVmax (HR 1.2; 95% CI [1.05–1.42]; *p* < 0.01) and liver lesion progression (HR 19.7; 95% CI [1.70–227.63]; *p* < 0.05) were kept in the final model. These data are presented in Table 8. In the multivariate stepwise Cox analysis of 5-year OS, a significant model combining mediastinum lesion progression, lung SUVmax, liver lesion progression and neck lesion progression was found (*p* < 0.05) while excluding the other parameters. These data are presented in Table 9.

## 4. Discussion

In this study, we showed that ^18^FDG PET/CT metrics of metastasis located in the mediastinum and the liver and of the overall tumor burden were prognostic factors of poor overall survival at 1 year and 5 years. 

The population of our study was similar to most studies on RAI-R DTC in age and sex distribution, but our patients showed a higher frequency of aggressive pathologic types, at 32%, while the usual rate is 3–24% [35,36,37]. Consistently, the median OS of our population was 22 months, shorter than the expected 3–5 years described in the literature [38]. On the contrary, in our population, only 20.5% of patients had stage IV AJCC/TNM while this group usually represents 40–50% of patients with RAI-R DTC. The reason for this difference in staging is that we used the eighth edition of the AJCC/TNM classification, implemented on 1 January 2018, which reclassified all metastatic patients aged between 45 and 55 years from stage IV to stage II [39]. These observations might have introduced a bias in our study since histologic type and AJCC/TNM stage are known prognostic factors of DTC survival. Nevertheless, age, sex, histologic type and stage were not significant prognostic factors of OS in this study. This might be explained by the fact that the inclusion criteria of our study selected patients at a high risk of disease-related death, thus abolishing the pertinence of usual risk factors.

The ^18^FDG PET/CT metrics we analyzed, MTV and SUVmax, were highly correlated by Spearman’s correlation, with a correlation coefficient close to 1 (minimal ρ = 0.74) regardless of the site of metastasis. We chose to study them nonetheless since both MTV and SUVmax have shown their prognostic value in previous studies [23,28].

Metabolic tumor volume is representative of tumor burden, and a high overall tumor burden has long been known to be a factor in poor prognosis. In 2006, a study by Durante et al. of 444 patients with distant metastasis showed that those who had macronodular lung metastasis or multiple bone metastases or both bones and lung metastases had a 7.3 relative risk of death (*p* < 0.0001) compared to patients with metastasis on 131 I-TBS but with normal chest and bone x-rays [7]. In another study in 2011 of 80 patients with metastatic DTC, the two-year survival was 50% for patients with more than 10 lesions with FDG uptake (*p* = 0.009) against 70-80% for patients with 1–10 lesions and 100% for patients without lesions with FDG uptake [24]. Furthermore, in a recent study of 717 patients with metastatic DTC, the presence of 3 or more different distant organ system metastases was the only independent prognostic factor of 10-year OS by multivariate analysis [25]. Concordantly, Terroir et al. showed that an MTV/patient ≥ 15.2 cm^3^ was predictive of poorer 1-year and 2-year OS (*p* = 0.005) [29] and Manohar et al. showed that MTV values above the median were predictive of poorer progression-free survival (*p* = 0.007) [36]. Our results are therefore in accordance with previous studies.

The standardized uptake value is representative of the glycolytic rate of tumors, which is higher in malignant cells. It has been demonstrated that the majority of RAI-R lesions with high FDG avidity are of an aggressive histologic type and correspond to a transformation to a higher grade compared to the primary tumor [26]. Several studies have shown that the SUV is negatively correlated with OS and have suggested cut-off values ranging from 5 to 13.3 g/mL [23,24,27,28]. In our study, SUVmax cut-off values were 3.9 for the mediastinum (*p* = 0.002), 5.3 for the liver (*p* = 0.001) and 18.6 for the total SUVmax (*p* = 0.026).

The most common site of metastasis of DTC is the lungs [3,7,10,24,40], a finding which was also true in our study with 71% of patients bearing lung metastasis. Respiratory deficiency is the main cause of death in patients with metastatic DTC [41,42]. However, lung MTV was not found to be a significant prognostic factor either for 1-year OS or 5-year OS in this study. This could be explained by the frequency of miliary metastases for which the estimation of tumor volume is challenging on ^18^FDG PET/CT and was observed in 72% of our patients who had lung metastasis. Similarly, a study of 138 patients with metastatic DTC did not find that lung tumor volume was predictive of survival, despite a significant effect on progression-free survival in univariate and multivariate analysis [43].

Liver MTV was also a significant prognostic factor identified in our study, most probably because metastases in this organ attest to a more advanced stage of illness. Indeed, liver metastases occur in less than 7% of cases of metastatic DTC that are not RAI-R [7,13,35]. On the contrary, in our study, 24% of patients had liver metastases, a rate comparable to those described in the literature for RAI-R DTC that range from 16 to 36% [25,44,45,46] and liver metastases are rarely sensitive to RAI treatment [47]. The median volume of liver metastasis in our study was zero, with a range of 0–181.2 cm^3^, which means that the results obtained from the analysis of these data can be interpreted as binary instead of continuous. This was evidenced by the binary univariate Cox analysis, which showed a significant association between liver MTV (HR 15.2; *p* = 0.015) and overall survival at 1 year. Shah et al. analyzed a cohort of 11 patients bearing liver metastases from DTC and showed that the survival rate was poor but could not be attributed to liver metastases per se because of the extensive metastatic disease at other sites [48]. Therefore, the presence of liver metastasis itself is a factor of a poor prognosis, regardless of volume.

Patients demonstrating more rapid disease progression over time can be expected to reach a lethal tumor burden more quickly than patients with slowly growing or stable disease [49]. In our study, this was only true for the progression of lesions in the mediastinum and the liver, as they were the only sites of the progression predictive of poor 1-year and 5-year OS. As discussed above, the progression of pulmonary lesions may have been underestimated because of resolution limitations. Of the 24 patients who had neck lesions, 12 (50%) underwent surgery (*n* = 9) or external beam therapy (EBT) (*n* = 3) after the reference PET/CT. The absence of correlation between the progression of neck lesions and OS could be explained by the efficacy of surgical reintervention and EBT. The progression of bone metastases might not be predictive of poor OS because they are associated with skeletal-related events responsible for functional-threatening rather than life-threatening complications and usually respond to local treatments such as EBT, radiofrequency or cementoplasty. Therefore, our study reveals the importance of providing more effective treatment for liver and mediastinum lesions that may have fewer therapeutic resources than neck or bone lesions.

In our study, death occurred in 59% of patients. The causes of death were respiratory deficiency (40%), heart failure (10%), kidney failure (5%) or unknown (45%). Thyroid cancer is the main cause of death in patients with metastatic DTC [41,50,51,52]. Respiratory insufficiency due to large pulmonary metastases replacing lung tissue, massive hemorrhage and airway obstruction due to uncontrolled local tumors, and circulatory failure resulting from compression of the vena cava by extensive mediastinal or sternal metastases have been found to be the most important immediate causes of death [41,42,53]. These findings are consistent with the results of our study, which demonstrated that neck MTV was a prognostic factor of 1-year and of 5-year OS and that mediastinum MTV was a significant factor of poor prognosis by ROC and Kaplan–Meier analysis (*p* = 0.002).

One of the strengths of our study is that we have analyzed a rare population, whose incidence is estimated at 4–5 cases per million inhabitants [5]. Indeed, our enrolment of 34 patients is comparable to other cohorts with similar inclusion criteria [27,29,35,36,37].

Our primary endpoint was overall survival at 1 year and 5 years. Overall survival appears to be the most relevant endpoint because it is a precise, objective event not subject to observer interpretation, unlike progression-free survival or disease-specific survival. The median overall survival of patients with iodine-refractory CDT is 3 years [54] so we chose to study overall survival at 1 year and 5 years to identify short- and medium-term factors of poor prognosis.

One of the main strengths of our study is its originality. Many authors have focused on FDG PET/CT parameters in search of prognostic factors of survival in RAI-R DTC. In contrast, this study is the only one to our knowledge that has investigated these parameters in relation to the organs bearing DTC metastasis. This approach allows a better correlation of FDG PET/CT parameters with the immediate causes of death, thus identifying in patients the lesions at risk of poor prognosis that would require early management.

The main weakness of our study was its retrospective and single-center nature, which included a small number of patients. As stated above, RAI-R DTC is a rare disease, the diagnosis of which is based on a range of investigations (thyroglobulin level, anatomical imaging, post-therapy scintigraphy, FDG PET/CT). For these reasons, the implementation of a prospective study to identify prognostic factors for survival is difficult in this population.

Among the FDG PET/CT parameters, we did not study the SUL peak (peak SUV normalized to the patient’s lean mass). This is the recommended quantitative component according to the PERCIST (PET Response Criteria In Solid Tumors) criteria used for the follow-up of patients with solid cancers. The SUL peak was a prognostic factor of progression-free survival in one study [27]. It was not possible for us to analyze this parameter because of the heterogeneity of the CT scans to which PET was coupled and from which lean mass was calculated.

This study included patients managed at the Henri Becquerel Cancer Centre between April 2007 and December 2019. During this period, two multikinase inhibitor treatments, sorafenib and lenvatinib, were approved by the European Medicines Agency in 2015. The absence of systemic treatment prior to FDG PET/CT was among our inclusion criteria. However, some patients, particularly those included after 2015, were able to benefit from these new therapies, which have demonstrated efficacy on survival [12,13]. This difference in management could have introduced a bias in our study on survival between patients who were treated or not with multikinase inhibitors. However, in our study, kinase inhibitor treatment was not correlated with OS at 1 year and at 5 years in Cox univariate analysis. This result does not refute the benefit of TKI treatment since our study was not designed for this purpose.

It has been shown that the sensitivity of FDG PET/CT may be increased for the number of lesions detected after recombinant human TSH (rhTSH) stimulation compared to FDG-PET/CT performed on suppressive thyroid hormone treatment [55]. However, the sensitivity for detecting patients with at least one tumor site was not improved by the rhTSH stimulation and the clinical benefit of identifying these additional small lesions remains to be proven.

Finally, the presence of the BRAF^V600E^ mutation and thyroglobulin doubling time <1 year are poor prognostic factors that we did not include in our analysis. A recent study confirmed that thyroglobulin correlated with overall survival but that MTV was not significantly correlated with thyroglobulin levels [56]. The retrospective design of our study did not allow for the reliable collection of thyroglobulin levels and BRAFV600E mutation of the included patients because blood samples were often analyzed in external laboratories and surgery was frequently conducted in a care center other than the Henri Becquerel Cancer Centre.

In conclusion, in this study, we demonstrated that in radio-iodine refractory differentiated thyroid cancer, ^18^FDG PET/CT parameters of the mediastinum, the liver and the overall tumor burden were prognostic factors of poor overall survival at 1 year and 5 years. The identification of these criteria could allow patients to benefit from an early therapeutic intervention in order to improve their overall survival.

## Figures and Tables

**Figure 1 diagnostics-12-02381-f001:**
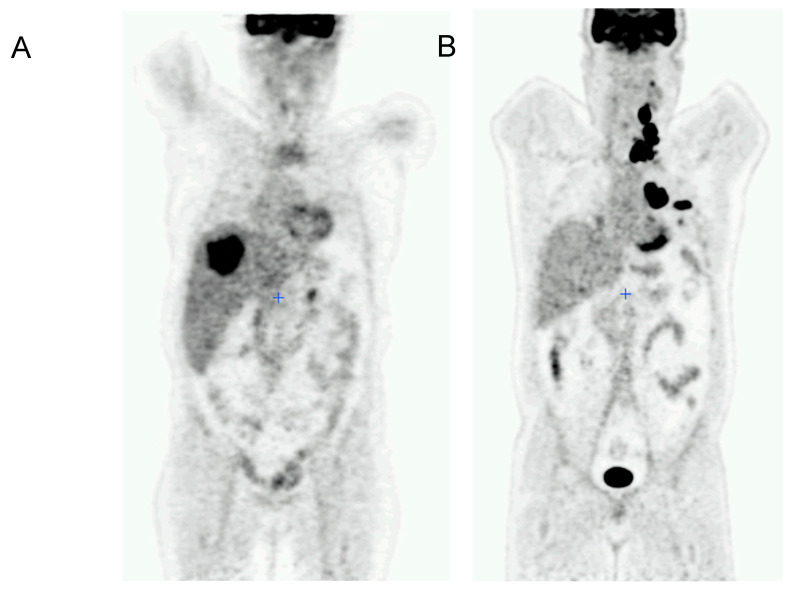
18F-FDG-PET/CT acquisitions of (**A**) a 53-year-old woman, liver MTV is 181 cm^3^, and (**B**) of a 51-year-old man, mediastinum SUVmax is 42.2 g/mL.

**Table 1 diagnostics-12-02381-t001:** Patients clinical characteristics for the whole population (*n* = 34).

Patients Characteristics		Whole Population (*n* = 34)
Sex	Female	19 (56%)
	Male	15 (44%)
Age (years) at diagnosis of RAI-R	<55	7 (21%)
	≥55	27 (79%)
Pathology		
**Papillary**		14 (42%)
	Classic	12 (86%)
	Tall cell variant	1 (7%)
	Columnar	1 (7%)
**Follicular**		9 (26%)
	Classic	7 (21%)
	Oncocytic	2 (12.5%)
**Poorly differentiated**		11 (32%)
Margin resection status	R0	23 (68%)
	R1	11 (32%)
Vascular invasion	Absent	14 (41%)
	Present	20 (59%)
Stage	I	8 (24%)
	II	17 (50%)
	III	2 (5.5%)
	IVa	2 (5.5%)
	IVb	5 (15%)
Cumulative RAI dose (mCi)	Med (SD)	200 (SD 89)
RAI-R inclusion criteria	“no RAI uptake”	27 (79%)
	“disease progression despite RAI avidity”	5 (15%)
	“extensive RAI exposure”	2 (6%)
Sites of metastasis	Neck	24 (71%)
	Mediastinum	13 (38%)
	Lung	24 (71%)
	Liver	8 (24%)
	Bone	13 (38%)
	Other	5 (15%)
Progressive lesion	Neck	13 (54%)
	Mediastinum	6 (46%)
	Lung	9 (38%)
	Liver	6 (75%)
	Bone	7 (54%)
	Other	3 (60%)
Tyrosine kinase inhibitor	Lenvatinib	1 (3%)
	Sorafenib	1 (3%)
	Pazopanib	2 (6%)
Death		20 (59%)
Delay to death (months)	Med (SD)	22 (SD 23)
Cause of death	Respiratory deficiency	8 (40%)
	Heart failure	2 (10%)
	Kidney failure	1 (5%)
	Unknown	9 (45%)

**Table 2 diagnostics-12-02381-t002:** Description of PET metrics on diagnostic ^18^FDG/PET-CT.

		Neck	Mediastinum	Liver	Bone	Lungs	Total
MTV	Mean	20.84	13.73	14.01	7.60	8.34	64.5
	*Median* ± SD	*1.11* ± 39.71	*0* ± 42.71	*0* ± 40.24	*0* ± 14.45	*0* ± 18.32	*24.12* ± 100.7
	[min–max]	[0–152]	[0–240]	[0–181]	[0–59.1]	[0–69.3]	[0–495]
SUVmax	Mean	13.02	6.35	3.93	5.50	5.35	16.7
	*Median* ± SD	*5.53* ± 16.6	*0* ± 11.2	*0* ± 10.2	*0* ± 9.76	*2.69* ± 7.95	*8.04* ± 16.9
	[min–max]	[0–56.7]	[0–42.2]	[0–42.1]	[0–36.9]	[0–32.1]	[0–56.7]

**Table 3 diagnostics-12-02381-t003:** Diagnostic performance of parameters measured on 18FDG PET/CT for 1-year overall survival using a ROC analysis. AUC: area under the curve, Se: sensitivity, Sp: specificity, Acc: accuracy.

	MTV						SUVmax						Progression					
	Threshold	AUC	Se	Sp	Acc	*p*-Value	Threshold	AUC	Se	Sp	Acc	*p*-Value	Threshold	AUC	Se	Sp	Acc	*p*-Value
Neck	11.73	0.724	0.8	0.759	0.882	0.057	18.61	0.752	0.8	0.759	0.882	0.037	1	0.51	0.4	0.621	0.853	0.477
Mediastinum	3.21	0.869	0.8	0.828	0.882	0.002	3.9	0.862	1	0.759	0.882	0.002	1	0.748	0.6	0.897	0.853	0.004
Liver	1.16	0.824	0.8	0.862	0.882	0.001	5.3	0.824	0.8	0.897	0.882	0.001	1	0.748	0.6	0.897	0.853	0.004
Bone	1.45	0.614	0.6	0.655	0.853	0.187	4.76	0.593	0.6	0.655	0.853	0.235	1	0.614	0.4	0.828	0.853	0.133
Lungs	19.39	0.583	0.4	0.897	0.853	0.244	3.94	0.724	0.8	0.724	0.882	0.055	1	0.462	0.2	0.724	0.853	0.649
Total	79.50	0.876	1	0.793	0.853	0.004	18.61	0.779	0.8	0.665	0.882	0.026	1	0.507	0.6	0.414	0.853	0.489

**Table 4 diagnostics-12-02381-t004:** Diagnostic performance of parameters measured on 18FDG PET/CT for 5-year overall survival using a ROC analysis. AUC: area under the curve, Se: sensitivity, Sp: specificity, Acc: accuracy.

	MTV						SUVmax						Progression					
	Threshold	AUC	Se	Sp	Acc	*p*-Value	Threshold	AUC	Se	Sp	Acc	*p*-Value	Threshold	AUC	Se	Sp	Acc	*p*-Value
Neck	1.65	0.729	0.667	0.75	0.706	0.011	6.33	0.747	0.722	0.875	0.794	0.007	1	0.448	0.333	0.562	0.529	0.737
Mediastinum	0.92	0.795	0.667	0.938	0.794	<0.001	3.79	0.792	0.667	0.938	0.794	<0.001	1	0.667	0.333	1	0.647	0.007
Liver	1.16	0.722	0.444	1	0.706	0.002	4.76	0.722	0.444	1	0.706	0.002	1	0.667	0.333	1	0.647	0.007
Bone	1.85	0.674	0.556	0.812	0.676	0.025	4.76	0.691	0.556	0.812	0.676	0.016	1	0.635	0.333	0.938	0.618	0.029
Lungs	7.23	0.606	0.389	0.875	0.618	0.099	3.04	0.757	0.667	0.812	0.735	0.005	1	0.514	0.278	0.75	0.529	0.437
Total	3.2	0.816	0.889	0.688	0.794	0.001	7.36	0.809	0.889	0.75	0.824	0.001	1	0.524	0.611	0.438	0.529	0.396

**Table 5 diagnostics-12-02381-t005:** Univariate Cox analysis of 1-year OS. HR: hazard ratio.

1-Year OS									
	MTV			SUVmax			Progression		
	HR	95% CI	*p*-Value	HR	95% CI	*p*-Value	HR	95% CI	*p*-Value
Neck	4.27	[0.98–18.56]	0.053	1.06	[1.01–1.12]	0.019	1.03	[0.17–6.14]	0.978
Mediastinum	1.57	[0.42–5.83]	0.504	1.08	[1.02–1.15]	0.014	8.25	[1.37–49.7]	0.02
Liver	2.55	[0.71–9.21]	0.152	1.04	[0.99–1.1]	0.093	8.25	[1.37–49.7]	0.02
Bone	3.80	[0.03–564.1]	0.601	0.99	[0.89–1.09]	0.815	2.57	[0.43–15.4]	0.301
Lungs	4.47	[0.11–175.4]	0.423	1.08	[1–1.16]	0.049	0.68	[0.08–6.09]	0.731
Total	1.55	[0.95–2.53]	0.077	1.06	[1–1.12]	0.042	1.03	[0.17–6.14]	0.977

**Table 6 diagnostics-12-02381-t006:** Univariate Cox analysis of 5-year OS. HR: hazard ratio.

5-Year OS									
	MTV			SUVmax			Progression		
	HR	95% CI	*p*-Value	HR	95% CI	*p*-Value	HR	95% CI	*p*-Value
Neck	2.77	[1.16–6.62]	0.02	1.04	[1.02–1.07]	0.002	0.74	[0.28–1.99]	0.554
Mediastinum	1.53	[0.77–3.02]	0.226	1.06	[1.02–1.10]	0.003	6.87	[2.35–20.06]	<0.001
Liver	2.56	[1.13–5.82]	0.0251	1.04	[1.01–1.08]	0.016	6.87	[2.26–20.82]	<0.001
Bone	3.97	[0.30–51.87]	0.294	1.04	[1–1.08]	0.052	2.93	[1.07–8.03]	0.036
Lungs	4.56	[0.53–39.11]	0.166	1.09	[1.04–1.15]	0.0004	0.97	[0.34–2.73]	0.952
Total	1.41	[1.07–1.86]	0.0155	1.05	[1.02–1.08]	0.001	1.05	[0.41–2.72]	0.915

**Table 7 diagnostics-12-02381-t007:** Multivariate Cox analysis of 1-year OS. HR: hazard ratio.

	MTV			SUVmax		
	HR	95% CI	*p*-Value	HR	95% CI	*p*-Value
Neck	1.04	[1.01–1.07]	0.0153	1.04	[0.96–1.11]	0.3453
Mediastinum	0.97	[0.94–1.01]	0.1398	1.22	[1–1.49]	0.0544
Liver	1.02	[1–1.04]	0.0418	1.07	[1–1.16]	0.0841
Bone	1.05	[0.95–1.16]	0.3431	0.80	[0.62–1.03]	0.0836
Lungs	1.01	[0.96–1.06]	0.7364	0.97	[0.78–1.22]	0.8021

**Table 8 diagnostics-12-02381-t008:** Multivariate stepwise Cox analysis of 1-year OS. HR: hazard ratio.

1-Year OS			
	HR	95% CI	*p*-Value
SUVmax–mediastinum	1.2	[1.05–1.42]	0.009
Progression–liver	19.7	[1.70–227.63]	0.017

**Table 9 diagnostics-12-02381-t009:** Multivariate stepwise Cox analysis of 5-year OS. HR: hazard ratio.

5-Year OS			
	HR	95% CI	*p*-Value
Progression–mediastinum	40.9	[5.48–305.30]	<0.001
SUVmax–lung	1.1	[1.03–1.19]	0.008
Progression–liver	8.9	[2.19–36.21]	0.002
Progression–neck	0.14	[0.03–0.70]	0.017

## Data Availability

Some or all datasets generated during and/or analyzed during the current study are not publicly available but are available from the corresponding author on reasonable request.

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
