# Peer review of "Using 18F-FDG-PET/CT Metrics to Predict Survival in Ra-Dio-Iodine Refractory Thyroid Cancers"

_diagnostics, 2022, doi:10.3390/diagnostics12102381_

Round 1
Reviewer 1 Report
In this retrospective study, Authors analyze the predictive value of 18FDG PET/CT about the overall survival in patients with RAI-R DTC. The topic is of interest because of its rarity.
Some observations:
1- Table 1: how many patients have multiple sites of metastases?
2- Table 1: Why only 3 pts were on TKI treatment?
3- 9 pts died for unknown causes: I don’t think is correct to include these patients in the analysis
4- Authors did not report or comment the hormonal data: what’s about thyreoglobulin and TH autoantibodies levels? Is there any correlation with 18FDG PET/CT data?
Author Response
Dear Reviewer 1,
Thank you very much for the interest you have shown for our study and for your accurate comments.
1- Table 1: how many patients have multiple sites of metastases?
Amongst our population of study of 34 patients, 23 patients had multiple sites of metastases. This 68% rate is superior to the 43% described in the study by Dineen et al. but this study comprised all metastatic DTC, and we can assume that RAI-R DTC would show a higher tumor burden as refractoriness usually appears at a more advanced stage of illness. Unfortunately, the rate of multiple sites of metastases is not described in the studies that have focused on RAI-R DTCs only.
2- Table 1: Why only 3 pts were on TKI treatment?
The 3 patients on TKI treatment mentioned are those for whom treatment was initiated during the 12 months that followed diagnosis of refractoriness, and consequently whose outcome in our study (1-year overall survival) could have been biased by TKI.
In total 13 patients received TKI treatment but in most cases (10 patients), TKI were initiated more than 12 months after diagnosis of RAI-R. Other patients were treated by local therapy (surgery, external beam radiation, radiofrequency etc.) as recommended.
3- 9 pts died for unknown causes: I don’t think it is correct to include these patients in the analysis.
In our study, 20 patients died, 9 of which the cause of death is unknown. Even though the specifics of the cause of death are unknown, we do think it is still interesting to include these patients in our study, because other authors have demonstrated that thyroid cancer is the main cause of death in patients with metastatic DTC.
4- Authors did not report or comment the hormonal data: what’s about thyroglobulin and TH autoantibodies levels? Is there any correlation with 18FDG PET/CT data?
This is a very valid point. A meta-analysis published in 2019 in the European Journal of Endocrinology showed that on patient-based analysis, 18F-FDG-PET/CT has high diagnostic accuracy for the detection of recurrent and/or metastatic diseases in DTC patients with thyroglobulin elevation. A recent study confirmed that thyroglobulin correlated with overall survival, but that TLG and MTV were not significantly correlated with thyroglobulin levels (Gay et al. 2-[18F]FDG PET in the Management of Radioiodine Refractory Differentiated Thyroid Cancer in the Era of Thyrosin-Kinases Inhibitors: A Real-Life Retrospective Study. Diagnostics (Basel). 2022 Feb 16;12(2):506).
However, we unfortunately could not include this factor in our analysis because the retrospective design of our study did not allow for reliable collection of thyroglobulin levels, blood samples being often taken in external laboratories.
This aspect has been added to the discussion.
In conclusion, we are truly grateful for your pertinent criticism which has enabled us to enrich our article.
We sincerely hope that the additions and modifications we have made to our study will fit your expectations and the standards of the journal.
Regards,
Authors

Reviewer 2 Report
This is a retrospective analysis of patients with radioiodine refractory thyroid carcinoma who had FDG PET/CT. FDG PET metrics were evaluated for their ability to predict 1- and 5-year OS in a group of 34 patients. The authors found FDG PET metrics of the mediastinal, liver, and whole-body tumor burden to be associated with poor 1- and 5-year OS.
The study is interesting and would make a useful contribution to the literature. Despite this, there are some important limitations to the work, including the heterogeneity of the disease included and the treatment offered to the patients.
Specific comments
1. Abstract – Results: Please include the 95%CI for the Hazard ratio obtained in Univariate and Multivariate Cox analyses.
2. Introduction – lines 43-44: Levothyroxine is rarely used in current practice as a first-line treatment option for thyroid carcinoma. Similarly, radioactive iodine is also not commonly applied as a first-line treatment option but is used as a follow-up treatment after an initial thyroidectomy. Please modify this sentence accordingly. Also, reference 6 cited here is a study comparing thyroxine withdrawal with recombinant TSH before radioactive iodine treatment of differentiated thyroid carcinoma and is not relevant here.
3. Introduction – lines 46-51: Please provide a reference for these definitions of RAI-R.
4. Methods – inclusion criteria: “(v) absence of systemic or local treatment before the reference 18FDG-PET/CT.” This inclusion criterion is ambiguous. Usually, patients would have been treated with radioactive iodine, which is a form of systemic treatment. Please clarify what is meant here.
5. Table 1: Please include the unit of the cumulative activity of I-131 administered for treatment.
6. Results: “Amongst known factors of poor prognosis, none were statistically significant in our study.” Which are the factors referred to here? Please show the results of these analyses.
7. Results: “These parameters were all highly correlated by Spearman’s correlation, with a correlation coefficient close to 1 (minimal ρ= 0.74) regardless of the site of metastasis.” Please include the numerical value for the correlation coefficients obtained for each parameter in table 2.
8. Results: “The MTV cut-off points were respectively 3.21 cm3 (AUC 0.869; p=0.002), 1.16 cm3 (AUC 0.824; p=0.001) and 79.5 cm3 (AUC 0.876; p=0.004).” Please include the body region of these MTV cut-offs.
9. Tables 5 and 6: Please include the 95% CI interval of all Hazard ratios computed.
1. Tables 7 and 8. Include the 95% CI of the HR. Please state the actual p-value in these tables as done in other tables.
1. Results: TLG, a product of MTV and SUVmean, is another FDG PET metric used for quantifying disease burden and has been shown to be a useful prognostic metric in different tumors, and in some studies, outperformed SUVmax and MTV for outcome prognostication. Why have the authors not included TLG in their analysis?
1. Discussion: FDG PET/CT was obtained while patients were on a suppressive dose of thyroxine. Some studies have reported improved detection rates with elevated TSH levels. It will be worthwhile that the authors to discuss the possible impact of thyroxine use on the quantified tumor burden in their patient population.
Author Response
Dear Reviewer 2,
Thank you very much for the interest you have shown for our study and for your accurate comments.
- Abstract – Results: Please include the 95%CI for the Hazard ratio obtained in Univariate and Multivariate Cox analyses.
Thank you for noticing this oversight. The results have been modified accordingly.
- Introduction – lines 43-44: Levothyroxine is rarely used in current practice as a first-line treatment option for thyroid carcinoma. Similarly, radioactive iodine is also not commonly applied as a first-line treatment option but is used as a follow-up treatment after an initial thyroidectomy. Please modify this sentence accordingly. Also, reference 6 cited here is a study comparing thyroxine withdrawal with recombinant TSH before radioactive iodine treatment of differentiated thyroid carcinoma and is not relevant here.
Thank you for pointing out this error, reference 6 has been modified to be replaced by the 2015 ATA recommendations (Haugen et al), as initially intended. The sentence describing treatment options for metastatic thyroid carcinoma has been modified accordingly.
3. Introduction – lines 46-51: Please provide a reference for these definitions of RAI-R.
The reference for definitions of RAI-R has been added (Schlumberger et al. Definition and management of radioactive iodine-refractory differentiated thyroid cancer. Lancet Diabetes Endocrinol. 2014;2(5):356–358.)
- Methods – inclusion criteria: “(v) absence of systemic or local treatment before the reference 18FDG-PET/CT.” This inclusion criterion is ambiguous. Usually, patients would have been treated with radioactive iodine, which is a form of systemic treatment. Please clarify what is meant here.
Thank you for noticing this issue that needs clarification. Indeed, patients need to have undergone treatment by radioactive iodine before being able to classify them as radioactive-iodine-refractory. What we meant here is absence of systemic treatment, but radioactive iodine, such as TKI. The sentence has been modified in the manuscript according to this comment.
- Table 1: Please include the unit of the cumulative activity of I-131 administered for treatment.
Thank you for observing this defect. The unit (mCi) of cumulative activity of I-131 administered for treatment has been added to table 1.
- Results: “Amongst known factors of poor prognosis, none were statistically significant in our study.” Which are the factors referred to here? Please show the results of these analyses.
Thank you for noticing this oversight. Known factors of poor prognosis, described in the introduction, are male gender, age ≥ to 55 years old, doubling time of thyroglobulin < 1-year, pathology type and the presence of BRAFV600Emutation. We were not able to assess doubling time of thyroglobulin and BRAF mutation because the retrospective design of our study did not allow for reliable collection of this data, blood samples and surgery being often conducted in other laboratories or medical centers. Results concerning age and histologic type have been added to the manuscript.
- Results: “These parameters were all highly correlated by Spearman’s correlation, with a correlation coefficient close to 1 (minimal ρ= 0.74) regardless of the site of metastasis.” Please include the numerical value for the correlation coefficients obtained for each parameter in table 2.
The correlation coefficient for each parameter ranged from ρ= 0.74 to ρ= 1.00, as shown if the figure below.
(cf document attached)
- Results: “The MTV cut-off points were respectively 3.21 cm3 (AUC 0.869; p=0.002), 1.16 cm3 (AUC 0.824; p=0.001) and 79.5 cm3 (AUC 0.876; p=0.004).” Please include the body region of these MTV cut-offs.
Thank you for pointing out this oversight. The sentence has been modified accordingly.
- Tables 5 and 6: Please include the 95% CI interval of all Hazard ratios computed.
Thank you for noticing this omission. The tables have been modified accordingly.
- Tables 7 and 8. Include the 95% CI of the HR. Please state the actual p-value in these tables as done in other tables.
Thank you for mentioning this lapse. The 95% CI of the HR have been added and p-values modified.
- Results: TLG, a product of MTV and SUVmean, is another FDG PET metric used for quantifying disease burden and has been shown to be a useful prognostic metric in different tumors, and in some studies, outperformed SUVmax and MTV for outcome prognostication. Why have the authors not included TLG in their analysis?
Thank you for pointing out this issue. We did include TLG in the design of our study, but as shown in the correlation plot, all three parameters (SUVmax, MTV and TLG) were highly correlated. As very accurately described in your comment, TLG being the product of SUVmean and MTV, we chose not to show these results in order to avoid redundancy.
- Discussion: FDG PET/CT was obtained while patients were on a suppressive dose of thyroxine. Some studies have reported improved detection rates with elevated TSH levels. It will be worthwhile that the authors to discuss the possible impact of thyroxine use on the quantified tumor burden in their patient population.
Thank you for noting this very valid point. As wisely mentioned, the sensitivity of 18-FDG PET/CT for detecting the number of lesions can be increased after recombinant human TSH stimulation, compared to FDG-PET/CT performed on suppressive thyroid hormone treatment. However, the sensitivity for detecting patients with at least one tumor site was not improved by the rhTSH stimulation and the clinical benefit of identifying these additional small lesions remains to be proven.
This point of interest has been added to the discussion.
In conclusion, we are truly grateful for your pertinent criticism which has enabled us to enrich our article.
We sincerely hope that the additions and modifications we have made to our study will fit your expectations and the standards of the journal.
Regards,
Authors

Round 2
Reviewer 1 Report
Manuscript has been changed as suggested
Reviewer 2 Report
Thank you for your responses to my comments. I am satisfied by the robustness of the responses provided. I have no further comments.